# Risk factors for recurrent endometriosis after conservative surgery in a quaternary care center in southern Thailand

**Pongpan Ngernprom, Satit Klangsin* , Chitkasaem Suwanrath ,
Krantarat Peeyananjarassri**

Department of Obstetrics and Gynecology, Faculty of Medicine, Prince of Songkla University, Hat Yai,
Songkhla, Thailand

* bukungmaru@gmail.com

## Abstract

### Objectives

To determine the 2-year recurrence rate of endometriosis after conservative surgery and
the risk factors for recurrence.

### Methods

This study retrospectively analyzed women with endometriosis who underwent conservative
surgery and had at least a 2-year follow-up at a quaternary care hospital in southern Thailand from January 2000 to December 2019. Recurrent endometriosis was defined as either
presence of endometrioma with a diameter $\geq$ 2 cm for more than three consecutive menstrual cycles or relapse of pelvic pain with the same or higher visual analog scale (VAS)
score as before surgery. Multivariate logistic regression analysis was used to identify the
risk factors for recurrence.

### Results

The median (interquartile range [IQR]) age was 34 (29, 38) years in 362 cases and nearly
three-quarters (74.2%) were nulliparous. Cyclic pain was the most common clinical presentation (48.9%) and the median (IQR) VAS score of pelvic pain was 6 (5, 9). Ovarian cystectomy was the most common procedure (68.1%). The 2-year recurrent endometriosis rate
after conservative surgery was 23.2%, and the overall recurrence rate was 56.4%. The risk
factors of recurrence were preoperative moderate to severe pelvic pain (adjusted odds ratio
[aOR] 1.93; 95% confidence interval [CI], 1.12–3.34; $p = 0.017$), adhesiolysis/ablation/ovarian cystectomy without unilateral oophorectomy (aOR 2.71; 95% CI, 1.40–5.23; $p = 0.002$),
and duration of postoperative hormonal treatment < 24 months (aOR 10.58; 95% CI, 5.47–
20.47; $p < 0.001$).

Mediche e Chirugiche (DIMEC), Orsola Hospital,
ITALY

**Data Availability Statement:** All relevant data are
within the paper and its supporting information
files.

**Funding:** The author(s) received no specific funding for this work.

**Competing interests:** The authors have declared that no competing interests exist.

## Conclusion

The 2-year recurrence rate after conservative surgery for endometriosis was 23.2%. Preoperative moderate to severe pelvic pain, procedures except unilateral oophorectomy, and postoperative hormonal treatment < 24 months were risk factors for recurrence.

## Introduction

Endometriosis, which is defined as the appearance of an endometrial glands with stroma outside the endometrium, is a common benign gynecological disease that affects 10% of reproductive women [1, 2]. Uncommonly, it was found to vary between 2% and 5% in menopausal women [3]. Similar to malignancy, it is characterized as a progressive and invasive growth that is estrogen-dependent and recurs with a tendency to metastasize. It is definitively diagnosed by histologic confirmation or by a clinically acceptable diagnosis from laparoscopic observation of the typical lesions as superficial "power-burn" or black, dark-brown, or blue "gunshot" lesions [1]. The American Society for Reproductive Medicine classifies the severity of endometriosis as minimal (I), mild (II), moderated (III), and severe (IV) using the revised American Fertility Society (rAFS) score based on intraoperative findings [4]. Treatment options of endometriosis are medical therapy with analgesic or hormonal therapy and surgery. Conservative surgery is usually performed in fertile or young women, which includes adhesiolysis, ablation or excision of the lesion, ovarian cystectomy, and unilateral oophorectomy [5–7].

The recurrence rates of endometriosis after conservative surgery varied according to the definition of recurrence and were reported as 9–29% and 13–30% at 2-year and overall recurrence rates, respectively [8–11]. Recurrent endometriosis can interfere with the quality of life and lead to repetitive surgery, difficult surgery, and finally radical surgery. The various factors reported to predict the recurrence of endometriosis after surgery included younger age, nulliparity, overweight, sexual intercourse, abnormal menstruation, presence and duration of secondary dysmenorrhea, previous hormonal use, presence of nodularity, cervical displacement on pelvic examination, and high rAFS score [1, 8, 9, 12–18]. The recurrence rate and associated risk factors were different among studies due to the definition of recurrence, sample size, duration of follow-up, types of surgery, experience of the surgeon, and postoperative medical therapy [1, 8, 9, 12–19].

Songklanagarind Hospital is a quaternary care referral center for 14 provinces in southern Thailand with experience in treating endometriosis for more than 20 years and our previous study showed that the cumulative recurrence rates of pain after depot medroxyprogesterone acetate (DMPA) treatment at months 12, 24, 36, 48, and 60 were 18%, 28%, 41%, 46%, and 50%, respectively [20]. Long-term follow-up for the recurrence of endometriosis in women who undergo conservative surgery has not been evaluated. Therefore, we aimed to determine the 2-year recurrence rate of endometriosis after conservative surgery and the predictive factors for recurrence.

## Materials and methods

A retrospective study was conducted at Songklanagarind Hospital after protocol approval by the Human Research Ethics Committee of the Faculty of Medicine, Prince of Songkla University (REC.64-210-12-4). The study population included women with endometriosis who underwent conservative surgery between January 2000 and December 2019. Diagnosis was

confirmed by either pathologic examination or intraoperative visualization of a typical endometriosis lesion. The inclusion criteria included age 18–45 years, received conservative surgery for the first time, and at least a 2-year follow-up after surgery. Patients who had emergency surgery for endometriosis, received radiotherapy, chemotherapy, or tamoxifen during follow-up, and pregnancy at the time of surgery were excluded.

The medical records of all eligible women were reviewed. Data collection included age, body mass index (BMI), symptoms of dysmenorrhea, infertility, parity, vaginal bleeding patterns according to the FIGO recommendation of 2011 [21], previous hormonal treatment, size of endometrioma, location of cysts, presence of adenomyosis or leiomyoma, presence of deep endometriosis, type of conservative surgery, approach of surgery, operative time, intraoperative blood loss, rAFS score, postoperative hormonal treatment, postoperative pregnancy, and recurrence time. The size of an endometrioma was defined as the largest diameter of the cysts. Obliteration of the pouch of Douglas was defined as any adhesion in the pouch. Pain was rated on the basis of a 10-cm visual analog scale (VAS) score, and the intensity was divided into none (0), mild (1–4), moderate (5–7), and severe (8–10) [8].

The indications for conservative surgery in our routine practice are fertility desire, or the preservation of ovarian function and an ovarian endometrioma $\geq 4$ cm, or pelvic pain with medical treatment failure. The surgical procedures depended on the endometriotic lesion, malignancy concern, and the operators' preference. Conservative surgery was performed by fellows training in obstetrics and gynecology or gynecologic staff doctors. The surgical procedures were as follows:

1. Adhesiolysis: a procedure used to separate or remove adhesions using scissors, cauterization, and blunting.

2. Ablation: using an electrical current technique to destroy the endometriotic lesions.

3. Excision of endometriotic lesion: removing the endometriotic lesions such as endometriotic spot, bleb or deep infiltrative lesion using scissors or cauterization.

4. Ovarian cystectomy: a stripping technique used to remove the endometriotic cyst wall after the plane between the cyst wall and the ovarian tissue has been cleaved.

5. Unilateral oophorectomy: a surgery that removes all ovarian pathology while leaving the contralateral ovary intact.

After surgery, the patients were observed for recurrence every 3 months for 1–2 years, then every 6–12 months thereafter. Postoperative hormonal treatment was prescribed for some patients based on pregnancy desire and clinical judgement of the staff doctors. The recurrence of endometriosis was defined as either pain or endometrioma that recurred during follow-up. The pain recurrence was defined as relapse of pain with a VAS score of at least equal to the score before surgery for at least 3 months. Recurrent endometrioma was defined as the presence of a new ovarian cyst characterized by a thin wall with a diameter of at least 2 cm, regular margin, and homogenous low echogenic fluid content with scattered internal echoes that, persisted for 3 consecutive menstrual cycles [8].

## Statistics analysis

The statistical analysis was performed using R 4.1.3 (R Foundation for Statistical Computing, Vienna, Austria). The data were presented as absolute numbers and percentages for categorical variables and as the median and interquartile range (IQR) for continuous variables. A univariate analysis was initially performed to identify any potential predictor variables. Variables with

a p-value < 0.2 according to the univariate analysis and variables considered to be clinically relevant were included in a multivariate analysis to determine any independent predictors. A multivariate logistic regression analysis was used to determine the associations of the potential risk factors with the primary outcome variables and to estimate the adjusted odds ratios (aOR) and the 95% confidence intervals (CI). We used a variable inflation factor below 5 to avoid multicollinearity for a logistic regression model. A stepwise selection procedure was used to identify variables with a p-value < 0.05.

## Results

The records of 367 women who underwent conservative surgery for endometriosis were reviewed. After excluding 2 women due to chemotherapy and 3 women due to pregnancy, 362 women were included in the study. Table 1 and S1 Table show the demographic and clinical characteristics, surgical approaches, operative findings, and postoperative data of the recurrence and non-recurrence groups.

The most common clinical presentation was cyclic pain. Laparoscopy was the most common approach and 6.8% of cases were converted to laparotomy. The majority of conservative surgeries were adhesiolysis and/or ablation and/or ovarian cystectomy. Based on intraoperative findings, ovarian endometrioma was found in most cases.

Post-operative hormonal treatment was administered in 237 women: DMPA (154), continuous combined oral contraceptives (COCs) (58), cyclic COCs (49), gonadotropin releasing hormone (GnRH) agonists (49), dienogest (39), levonorgestrel-releasing intrauterine system (LNG-IUS) (11), and monthly injectable contraceptives (9). Some patients had more than one hormonal treatment.

Compared to the non-recurrence group, the recurrence group was younger, had a higher rate of infertility, longer duration of a second dysmenorrhea, higher rate of operations other than unilateral oophorectomy, absence of endometrioma, lower rate of severe stage by rAFS, and a lower rate of receiving post-operative hormonal treatment.

The cumulative recurrence rates of endometriosis after conservative surgery are shown in Fig 1. The two-year recurrence rate was 23.2% (84/362 cases), whereas the overall recurrence rate was 56.4% with a median (IQR) duration of recurrence of 49 months (42, 55). The earliest time of recurrence was 3 months after surgery. During the first 5 years, a rapidly increasing rate of recurrence was observed, then the rate of recurrence became slower and finally constant after 9 years. However, only 11 cases were followed up after 9 years.

Univariate analysis and Cox's multivariate proportional hazard analysis were performed (Table 2). From the univariate analysis, we chose 11 factors with a p-value < 0.2 to include in the multivariate logistic regression model: (i) age at surgery (≤ 35 vs. > 35 years), (ii) duration of dysmenorrhea (< 12 vs. ≥ 12 months), (iii) severity of preoperative pelvic pain (none to mild vs. moderate to severe), (iv) dyspareunia (no/yes), (v) infertility (no/yes), (vi) palpation of pelvic mass (no/yes), (vii) surgical approach (laparotomy vs. laparoscopy), (viii) adenomyosis (no/yes), (ix) procedure (unilateral oophorectomy with or without adhesiolysis/ablation/ovarian cystectomy vs. adhesiolysis/ablation/ovarian cystectomy), (x) rAFS stage of endometriosis (minimal to mild vs. moderate to severe), and (xi) duration of postoperative hormonal treatment (< 24 vs. ≥ 24 months). We did not choose the factors following endometrioma and size of largest endometrioma for the univariate analysis because those factors are the components of rAFS score calculation. We found three factors that were independently associated with recurrence: preoperative moderate to severe pelvic pain, surgical procedures other than unilateral oophorectomy (adhesiolysis/ablation/ovarian cystectomy), and duration of postoperative hormonal treatment < 24 months. Fig 2. compares the 3 groups of women who received no

**Table 1.** Demographic and clinical characteristics, surgical approaches, operative findings, and postoperative data in the recurrence and non-recurrence endometriosis groups.

| Variables | Total N (%) (N = 362) | Nonrecurrence n (%) (n = 158) | Recurrence n (%) (n = 204) | P value |
|---|---|---|---|---|
| **Demographic and clinical characteristics** | | | | |
| Age at surgery (year)[a] | 34 (29, 38) | 35 (30.2, 40) | 33 (29, 37) | 0.040 |
| Parity [a] | 0 (0,1) | 0 (0,1) | 0 (0, 1) | 0.981 |
| BMI (kg/m$^2$)[a] | 21.7 (19.4, 23.9) | 21.4 (19.5, 23.9) | 21.9 (19.2, 23.8) | 0.966 |
| Abnormal menstruation by FIGO 2011 | 36 (9.9) | 17 (10.8) | 19 (9.3) | 0.780 |
| Chief complaint | | | | 0.194 |
| Cyclic pain | 177 (48.9) | 69 (43.7) | 108 (52.9) | |
| Acyclic pain | 46 (12.7) | 25 (15.8) | 21 (10.3) | |
| Acute pelvic pain | 35 (9.7) | 21 (13.3) | 14 (6.9) | |
| Infertility | 31 (8.6) | 12 (7.6) | 19 (9.3) | |
| Pelvic mass | 28 (7.7) | 11 (7.0) | 17 (8.3) | |
| Others | 45 (12.4) | 20 (12.6) | 25 (12.3) | |
| Secondary dysmenorrhea | 285 (78.7) | 118 (74.7) | 167 (81.9) | 0.127 |
| Duration of secondary dysmenorrhea (month)[a] | 12 (6, 36) (n = 285) | 11 (5, 24) (n = 118) | 24 (7.5, 36) (n = 167) | < 0.001 |
| VAS score[a] | 6 (5, 9) | 6 (5, 8) | 6 (5, 9) | 0.106 |
| Preoperative moderate to severe pelvic pain (VAS $\geq$5) | 266 (73.5) | 108 (68.4) | 158 (77.5) | 0.068 |
| Dyspareunia | 39 (10.8) | 12 (7.6) | 27 (13.2) | 0.122 |
| Infertility | 113 (31.2) | 36 (22.8) | 77 (37.7) | 0.004 |
| Previous hormonal use | 93 (25.7) | 36 (22.8) | 57 (27.9) | 0.321 |
| Palpation of pelvic mass | 74/340 (21.8) | 44/148 (29.7) | 30/192 (15.6) | 0.003 |
| Cervical displacement | 80/104 (76.9) | 39/51 (76.5) | 41/53 (77.4) | 1.000 |
| Nodularity of cul-de-sac | 88/177 (49.7) | 36/74 (48.6) | 52/103 (50.5) | 0.929 |
| **Surgical approach and operative findings** | | | | |
| Laparoscopic approach | 266 (73.5) | 105 (66.5) | 161 (78.9) | 0.011 |
| Operative procedure[b] | | | | < 0.001 |
| Adhesiolysis/ ablation/ ovarian cystectomy | 299 (82.9) | 116 (73.4) | 183 (89.7) | |
| Unilateral oophorectomy with or without adhesiolysis/ ablation/ ovarian cystectomy | 63 (17.4) | 42 (26.6) | 21 (10.3) | |
| Obliterate cul-de-sac | 227/301 (75.4) | 104/136 (76.5) | 123/165 (74.6) | 0.394 |
| Uterosacral ligament involvement | 58/109 (53.2) | 21/46 (45.7) | 37/63 (58.7) | 0.247 |
| Deep endometriosis | 16/69 (23.2) | 6/31 (19.4) | 10/38 (26.3) | 0.693 |
| Endometrioma | 314 (86.7) | 148 (93.7) | 166 (81.4) | < 0.001 |
| Bilateral endometrioma | 125/314 (39.8) | 57/148 (38.5) | 68/166 (41) | 0.665 |
| Size of largest endometrioma (mm)[a] | 60 (50, 80) (n = 314) | 60 (50, 80) (n = 148) | 50 (40, 80) (n = 166) | 0.085 |
| rAFS score[a] | 48 (28, 87) (n = 283) | 48 (28, 87) (n = 126) | 60 (28, 92) (n = 157) | 0.895 |
| Severity/stage of rAFS score | (n = 283) | (n = 126) | (n = 157) | 0.017 |
| Minimal/ I | 13 (4.6) | 4 (3.2) | 9 (5.7) | |
| Mild/ II | 9 (3.2) | 1 (0.8) | 8 (5.1) | |
| Moderate/ III | 82 (29.0) | 46 (36.5) | 36 (22.9) | |
| Severe/ IV | 179 (63.2) | 75 (59.5) | 104 (66.3) | |
| Adenomyosis | 68 (18.8) | 36 (22.8) | 32 (15.7) | 0.114 |
| **Postoperative data** | | | | |
| Postoperative hormonal treatment | 237 (65.4) | 124 (78.4) | 113 (55.3) | < 0.001 |

*(Continued)*

**Table 1.** (Continued)

| Variables | Total N (%) (N = 362) | Nonrecurrence n (%) (n = 158) | Recurrence n (%) (n = 204) | P value |
|---|---|---|---|---|
| Duration of postoperative hormonal treatment (month[a] | 26 (13, 39.5) (n = 237) | 31 (24, 51) (n = 124) | 15 (8, 27) (n = 113) | < 0.001 |

BMI, body mass index; mm, millimeter; rAFS, revised American Fertility Society; VAS, visual analog scale.

[a]Data are presented as median (interquartile range).

[b]Some patients underwent surgery with more than one procedure.

Data were compared using Wilcoxon Rank Sum test for continuous data and the chi-square test or Fisher's exact test for categorical data.

P values of 0.05 were significant and analyzed between the non-recurrence and recurrence groups.

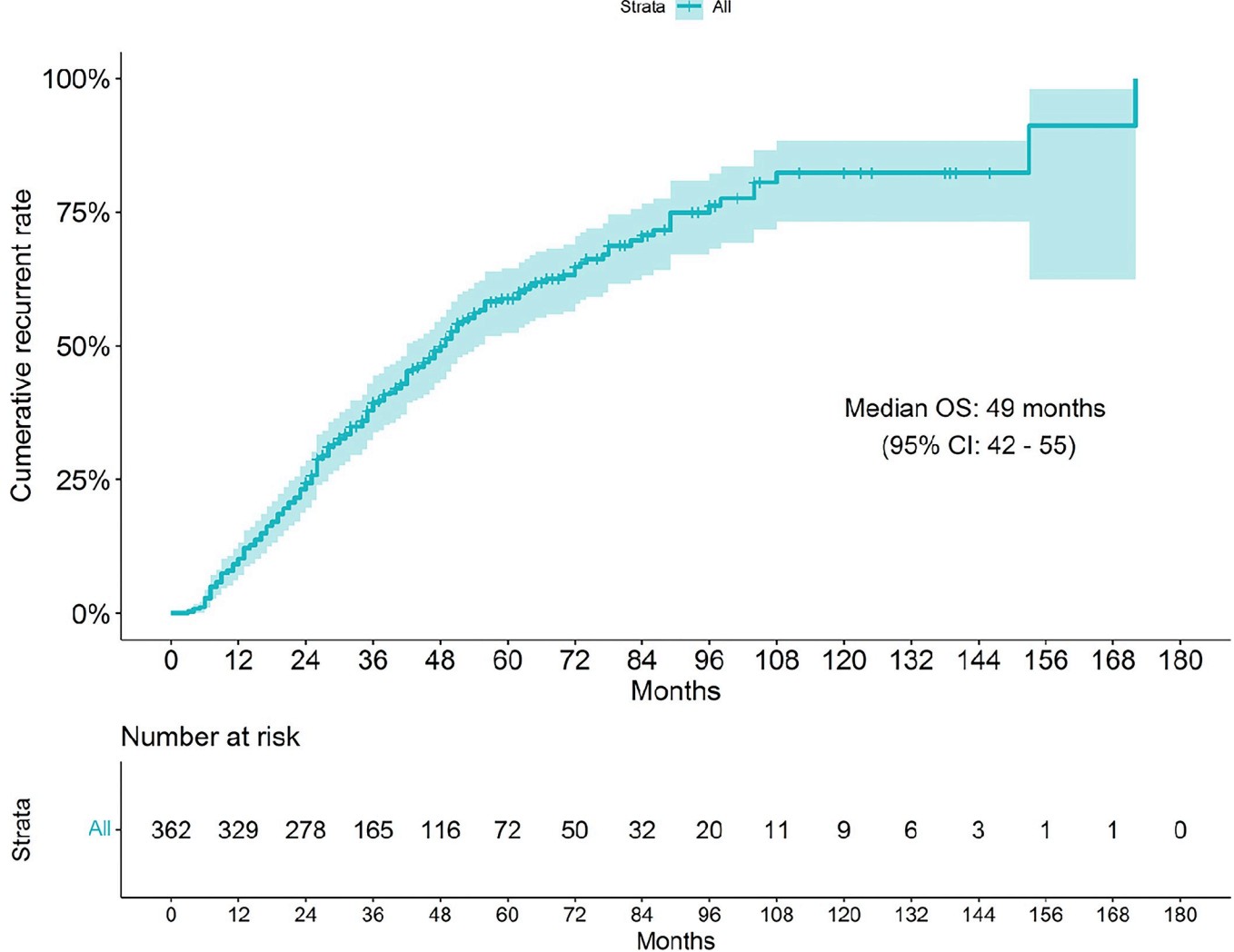

**Fig 1. Cumulative recurrence rates of endometriosis after conservative surgery (N = 362).**

**Table 2. Univariate and multivariate logistic regression analysis of risk factors for recurrent endometriosis after conservative surgery (n = 204).**

| Factors | Univariate analysis | | | Multivariate analysis | | |
|---|---|---|---|---|---|---|
| | OR | 95% CI | *P* value | OR | 95% CI | *P* value |
| Age at surgery ≤ 35 years old | 1.74 | 1.14–2.66 | 0.010 | 1.57 | 0.94–2.63 | 0.086 |
| Duration of dysmenorrhea ≥ 12 months | 2.22 | 1.28–3.84 | < 0.001 | 1.32 | 0.58–3.01 | 0.504 |
| Preoperative moderate to severe pelvic pain | 1.59 | 0.99–2.54 | 0.052 | 1.93 | 1.12–3.34 | 0.017 |
| Dyspareunia | 1.86 | 0.91–3.79 | 0.081 | 1.61 | 0.53–4.88 | 0.396 |
| Infertility | 2.02 | 1.27–3.23 | 0.003 | 1.31 | 0.70–2.42 | 0.397 |
| Palpation of pelvic mass | 1.92 | 0.98–3.76 | 0.002 | 1.78 | 0.87–3.62 | 0.111 |
| Laparoscopic approach | 1.89 | 1.18–3.03 | 0.008 | 1.36 | 0.74–2.51 | 0.319 |
| Absence of adenomyosis | 1.59 | 0.93–2.69 | 0.088 | 1.88 | 1–3.53 | 0.050 |
| Procedures other than unilateral oophorectomy | 3.16 | 1.78–5.60 | < 0.001 | 2.71 | 1.40–5.23 | 0.002 |
| Minimal to mild stage of endometriosis | 2.94 | 1.84–4.70 | < 0.001 | 1.43 | 0.42–4.8 | 0.560 |
| Duration of post-operative hormonal treatment < 24 months | 9.87 | 5.22–18.65 | < 0.001 | 10.58 | 5.47–20.47 | < 0.001 |

OR, odds ratio; 95% CI, 95% confidence interval.

Significant risk factors using multivariate logistic regression at a *p* value of 0.05.

postoperative hormonal treatment (91/125 [72.8%]), postoperative hormonal treatment < 24 months (69/86 [80.2%]), and postoperative hormonal treatment ≥ 24 months (44/151 [29.1%]) in terms of recurrence of endometriosis after conservative surgery. Only postoperative hormonal treatment ≥ 24 months had a significantly lower recurrence rate.

In women with recurrent endometriosis, lesion and pain recurrences were the most common clinical presentations (42.6%) followed by recurrence of pain (30.9%) and recurrence of lesion (26.5%). The most common symptom presentation was dysmenorrhea (63.7%). Treatments in recurrent endometriosis were medication (61.3%), surgery (38.2%), and follow-up (0.5%).

## Discussion

This study revealed that the 2-year recurrence rate and the overall recurrence rate of endometriosis after conservative surgery were high. By definition in this study, recurrent endometriosis is either a relapse of pelvic pain with the same or higher VAS score as before surgery or detection of a new endometrioma at least 2 cm in diameter that persisted for at least 3 consecutive menstrual cycles. The 2-year recurrence rate in this study was 23.2%, which was in the range from previous studies (9.2–29.4%) [9, 11, 12]. Nevertheless, the overall recurrence rate for the 12-year follow-up was as high as 56.4%, which was higher than previous studies (29.4–42%) [9, 10, 15]. However, the present study had a longer follow-up period than previous studies that had follow-up durations of 29–84 months [9, 10, 15]. Sengoku et al. (2013) [9] reported that the cumulative recurrence of endometriosis reached 42% at 60 months and defined recurrence as the presence of an ovarian cyst ≥ 2 cm in diameter without data on recurrent pelvic pain. A previous study reported recurrent endometrioma and pelvic pain at 28.7% and 33.4%, respectively [15]. Also, the present study had a higher overall recurrence than previous study conducted in Thailand (22.6%), which had a smaller sample size (n = 106) and shorter follow-up period of 1 year [22].

Based on the cumulative recurrent endometriosis in the present study, the rapid recurrence rate was observed in 60 months after surgery, then became slower after 60 months to 108 months, then constant after 108 months after surgery. This pattern of cumulative recurrence was similar to a previous study [9].

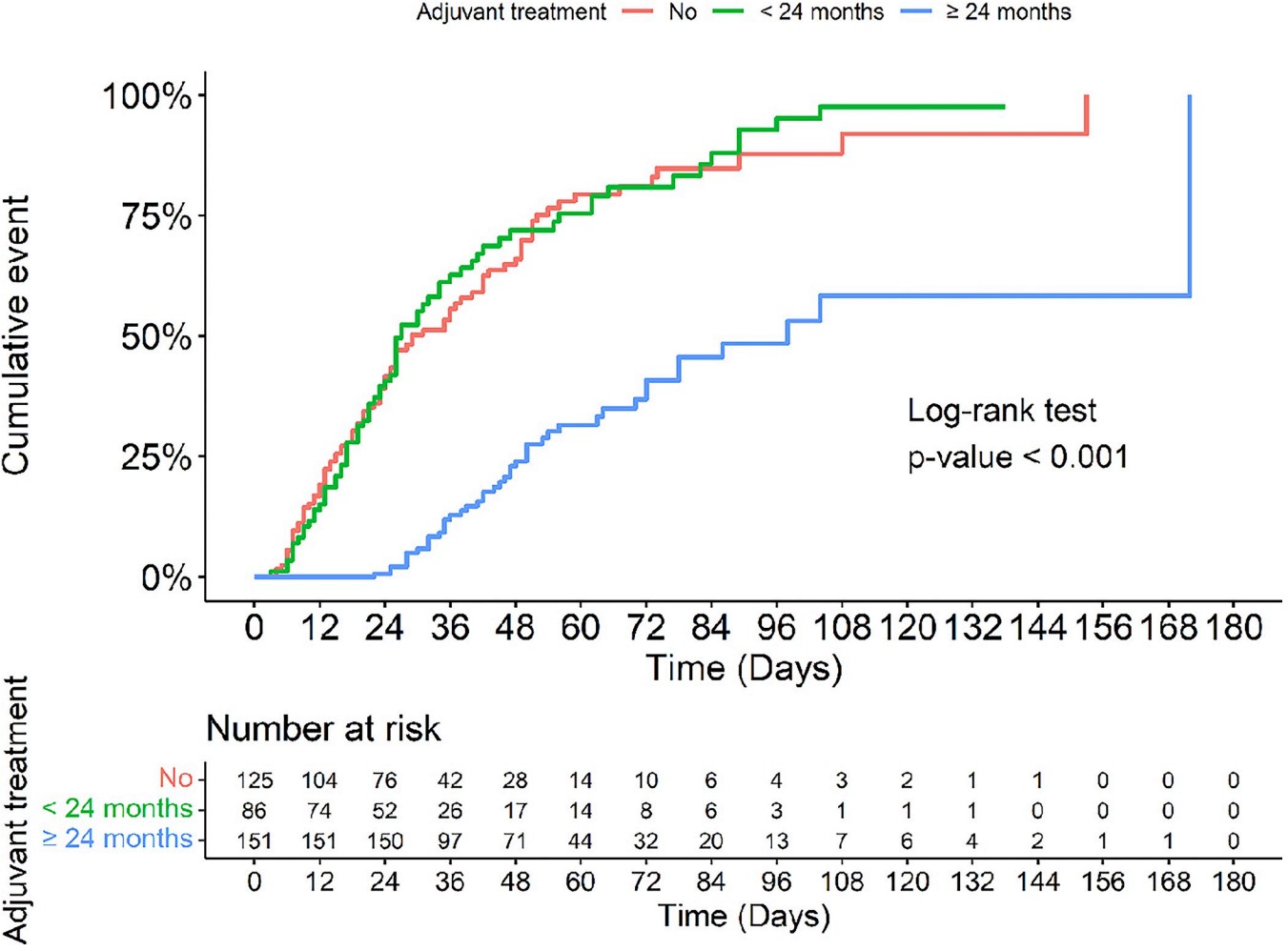

**Fig 2. Comparison of the hormonal treatment groups in terms of cumulative recurrence rates of endometriosis after conservative surgery: No treatment (n = 125), postoperative hormonal treatment < 24 months (n = 86), and postoperative treatment ≥ 24 months (n = 151).**

Factors that had a significantly associated risk for the recurrence of endometriosis after conservative surgery were preoperative moderate to severe pelvic pain, procedures other than unilateral oophorectomy (i.e., adhesiolysis/ablation/ovarian cystectomy), and duration of postoperative hormonal treatment < 24 months. Severity of pain and procedures other than unilateral oophorectomy were new predictive factors in this study, which have never been previously reported.

Endometriosis-associated pain has multiple pathogenesis, including inflammation and nociception from cyclic recurrent bleeding in endometriotic lesions, as well as a decrease in pain threshold in the peripheral and central nervous systems [23, 24]. Previous studies reported that preoperative dysmenorrhea was a significant risk factor for recurrence but lacked data on the severity of pain [17, 25]. This current study revealed more details on the severity of pelvic pain in that moderate to severe pelvic pain significantly increased the recurrence of endometriosis (aOR 1.93; 95% CI, 1.12–3.34; $p$ = 0.017). Previous studies showed severe pelvic pain was associated with complete pouch of Douglas obliteration and uterosacral ligament involvement, which were correlated with a high stage of endometriosis by rAFS score [26, 27]. Pluchino et al. (2020) [28] discovered a correlation between severity of symptoms and estrogen receptor-α levels.

The present study demonstrated that the risk factors for recurrent endometriosis were procedures for conservative surgery (i.e., adhesiolysis/ablation/ovarian cystectomy) other than unilateral oophorectomy. This can be explained by the fact that unilateral oophorectomy removes all endometriotic lesions. This factor was not observed in previous studies, possibly due to the definition of conservative surgery or lack of data available for analysis. A recent study reported that incomplete surgery was a risk factor for recurrence and explained that residual endometriotic cells during surgery were the cause of persistent and recurrent disease [1].

Among 362 women with endometriosis, 63 cases (17.4%) underwent unilateral oophorectomy. This group had a significantly higher median (IQR) age (39 [35, 41] vs. 33 [29, 37]), a higher median (IQR) parity (0 [0, 2] vs. 0 [0, 0]), and a higher percentage of non-fertility need (82.3% vs. 65.8%) compared to non-oophorectomy group. To avoid overtreatment, we recommend not performing unilateral oophorectomy, especially in young women, nulliparity, and have a greater need for fertility.

A systematic review and meta-analysis enrolled 2,137 participants in conservative surgery of endometriosis. Postoperative hormonal treatments that included COCs, progestin, androgen, LNG-IUS, and GnRH were studied and showed a significantly decreased recurrence of endometriosis [29]. The protective effects of hormonal treatment were: (i) ovulation inhibition, (ii) inactive endometrial cells and tissue, and (iii) reduction of re-implanted endometrial cells in peritoneal organs [29]. Our study found that postoperative hormonal treatment of less than 24 months is a risk of recurrence. According to the European Society of Human Reproduction and Embryology (ESHRE) guidelines, hormonal treatment should be prescribed for at least 18–24 months after surgery [30].

There were no significant differences in age at surgery, parity, BMI, sexual intercourse, abnormal menstruation, presence and duration of secondary dysmenorrhea, previous hormonal use, presence of nodularity, cervical displacement on pelvic examination, and high rAFS score, which were inconsistent with previous studies [1, 8, 9, 12–18]. These differences might be due to various factors such as ethnicity, study design, sample size, duration of follow up, definition of recurrence, surgical technique, and experience of the operators.

In the present study, endometrioma was the most common finding (86.7%), co-existing with the presence of obliterate cul-de-sac (75.4%), uterosacral ligament involvement (53.2%), and deep endometriosis (23.2%). Previous studies showed higher incidences of co-existing uterosacral ligament involvement (73.8%) and deep endometriosis (43.2%) [31, 32]. These discrepancies may be attributed to the study design, which had missing data (64% and 81% in uterosacral ligament involvement and deep endometriosis, respectively) and all of the operative finding scores based on rAFS classification. Although rAFS classification is accepted and correlates to surgical complexity, Enzian classification is suitable for defining deep endometriosis [33]. Ianieri et al. (2022) [34] demonstrated that parametrectomy significantly improved the symptoms of dyschezia, dysmenorrhea, dysuria, dyspareunia, and chronic pelvic pain but worsened female sexual dysfunction. The failure to detect deep endometriosis resulted in undertreatment and, ultimately, a lack of improvement in their symptoms.

Our study had a large sample size with long-term follow-up and also had various types and procedures for endometriosis, which reflected the real practice in a quaternary care center in southern Thailand, where the difficult and complicated endometriosis cases are referred. Moreover, we reported more details of pain as preoperative moderate to severe pelvic pain and more details of the procedures of conservative surgery as the risks for recurrence. The limitations of this study should also be addressed. Some data, such as the rAFS score or clinical details, may have been missed or under reported due to the retrospective nature of this study. In addition, our study found various conservative surgical methods depended on the operators.

Based on the findings in this study, we recommend close followed-up of patients for at least 5 years after conservative surgery because of the high recurrence rate and continue the follow-up thereafter. In the first 1 to 2-year period, frequent follow-up visits should be required every 3 months, then every 6 months until 5 years, then yearly. Patients with preoperative moderate to severe pelvic pain and patients who undergo conservative surgery without unilateral oophorectomy, such as adhesiolysis, excision, ablation, or cystectomy, should be monitored for recurrence. Finally, according to the ESHRE guidelines [30], all patients should receive hormonal treatment for at least 24 months after surgery if there are no contraindications.

## Conclusion

The two-year recurrence rate of endometriosis after conservative surgery was 23.2%, while the overall recurrence rate was 56.4%. Preoperative moderate to severe pelvic pain, procedures that include adhesiolysis/ablation/ovarian cystectomy, except unilateral oophorectomy, and a postoperative hormonal treatment duration less than 24 months were risk factors for recurrence.

## Supporting information

**S1 Table. Comparison of variables between non-recurrence and recurrence groups of conservative surgery in endometriosis.**
(DOCX)

## Acknowledgments

This present study was supported by the Faculty of Medicine at Prince of Songkla University. I would like to thank Glenn Shingledecker for English editing.

## Author Contributions

**Conceptualization:** Satit Klangsin.

**Data curation:** Pongpan Ngernprom, Satit Klangsin, Chitkasaem Suwanrath, Krantarat Peeyananjarassri.

**Formal analysis:** Pongpan Ngernprom, Satit Klangsin, Chitkasaem Suwanrath, Krantarat Peeyananjarassri.

**Investigation:** Pongpan Ngernprom, Satit Klangsin.

**Methodology:** Satit Klangsin, Chitkasaem Suwanrath, Krantarat Peeyananjarassri.

**Project administration:** Satit Klangsin.

**Supervision:** Satit Klangsin, Chitkasaem Suwanrath, Krantarat Peeyananjarassri.

**Writing – original draft:** Pongpan Ngernprom, Satit Klangsin.

**Writing – review & editing:** Pongpan Ngernprom, Satit Klangsin, Chitkasaem Suwanrath, Krantarat Peeyananjarassri.

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
