## [Decision Letter · Decision Letter 0]

11 Jul 2023

PONE-D-22-29483Risk factors for recurrent endometriosis after conservative surgery in a quaternary care center in southern ThailandPLOS ONE

Dear Dr. Klangsin,

Thank you for submitting your manuscript to PLOS ONE. After careful consideration, we feel that it has merit but does not fully meet PLOS ONE’s publication criteria as it currently stands. Therefore, we invite you to submit a revised version of the manuscript that addresses the points raised during the review process.

We look forward to receiving your revised manuscript.

Kind regards,

Diego Raimondo

Academic Editor

PLOS ONE

Journal Requirements:

If your study used hospital data,  please ensure that you have discussed whether all data/samples were fully anonymized before you accessed them and/or whether the IRB or ethics committee waived the requirement for informed consent. If patients provided informed written consent to have data/samples from their medical records used in research, please include this information.

**Additional Editor Comments:**

Please revise the manuscript according to Reviewers's suggestions.

Reviewers' comments:

Reviewer's Responses to Questions

**Comments to the Author**

1. Is the manuscript technically sound, and do the data support the conclusions?

Reviewer #1: Yes

Reviewer #2: Yes

2. Has the statistical analysis been performed appropriately and rigorously? 

Reviewer #1: Yes

Reviewer #2: Yes

3. Have the authors made all data underlying the findings in their manuscript fully available?

Reviewer #1: Yes

Reviewer #2: Yes

4. Is the manuscript presented in an intelligible fashion and written in standard English?

Reviewer #1: Yes

Reviewer #2: Yes

5. Review Comments to the Author

Reviewer #1: The manuscript by Ngernprom et al is an interesting report on 362 patients followed-up after surgery for endometriosis. The Authors identified the risk factors for recurrence, some of which were not identified in the previous literature.

However, being the only novelty the identification of previously unreported risks factors for recurrence, one of which may be in fact a limitation of the study (in fact, 17% of oophorectomies in a population with a mean age of 34 may be considered an overtreatment, if details on surgery are not provided), I am not sure that the paper meets the high standards required for a high impact journal.

Minor issues:

Line 3 (short title): factors, not factor; surgery for endometriosis, not surgery in endometriosis.

Line 31 (abstract): with the same or higher VAS; add "or higher".

Line 58 (intro): glands, not gland.

Line 69 (intro): add some reference on the general treatment of endometriomas, such as Chapron C et al "Management of ovarian endometriomas", Hum Reprod Update 2002, and Muzii L et al "Management of endometriomas", Semin Reprod Med 2017.

Line 81 (intro): too many references to attest the high endometriosis load of the Authors (refs 17, 18, 19). Select only one of the three. Ref 20 is on the other hand appropriate in the text.

Line 264 (discussion): close follow-up, not closed

Reviewer #2: Dear author, thank you for the opportunity to analyze this interesting study on such a widespread pathology.

However, I believe that some elements should be specified in the introduction:

- endometriosis can be responsible for symptoms even in the menopausal age, varying between 2 and 5% of cases [you can cite: M M Ianieri et al. Retroperitoneal endometriosis in postmenopausal woman causing deep vein thrombosis: case report and review of the literature]

- I suggest specifying that the recurrence rate of endometriosis also varies according to the definition of recurrence among the various authors

In the discussion session:

- advice to improve and enrich the discussion about the potential role of deep endometriosis on disease recurrence, particularly if radical surgery has not been performed. It would in fact be suggested to specify this aspect in the results or even in the tables. In particular, it would be interesting to know in your series the incidence of endometriosis of the parameters which can often be associated with ovarian endometriosis and whose non-treatment can be associated with persistence/recurrence of symptoms [Ianieri MM et al. Impact of nerve-sparing posterolateral parametrial excision for deep infiltrating endometriosis on postoperative bowel, urinary, and sexual function].

You cannot speak of ovarian recurrence without specifying the possible infiltration of the parameters which are anatomical structures adjacent to the ovaries

6. PLOS authors have the option to publish the peer review history of their article (what does this mean?). If published, this will include your full peer review and any attached files.

Reviewer #1: No

Reviewer #2: No

---

## [Author Response · Author response to Decision Letter 0]

14 Jul 2023

Reviewer #1: The manuscript by Ngernprom et al is an interesting report on 362 patients followed-up after surgery for endometriosis. The Authors identified the risk factors for recurrence, some of which were not identified in the previous literature.

However, being the only novelty the identification of previously unreported risks factors for recurrence, one of which may be in fact a limitation of the study (in fact, 17% of oophorectomies in a population with a mean age of 34 may be considered an overtreatment, if details on surgery are not provided), I am not sure that the paper meets the high standards required for a high impact journal.

Response: We understood your mention of oophorectomy in conservative surgery. Additionally, I analyzed the data based on the oophorectomy group compared to the non-oophorectomy group, and discussion was more in the scope with which you are concerned.

Previous manuscript: -

Revised manuscript: Line 243-248 …… “Among 362 women with endometriosis, 63 cases (17.4%) underwent unilateral oophorectomy. This group had a significantly higher median (IQR) age (39 [35, 41] vs. 33 [29, 37]), a higher median (IQR) parity (0 [0, 2] vs. 0 [0, 0]), and a higher percentage of non-fertility need (82.3% vs. 65.8%) compared to non-oophorectomy group. To avoid overtreatment, we recommend not performing unilateral oophorectomy, especially in young women, nulliparity, and have a greater need for fertility.”

Minor issues:

Line 3 (short title): factors, not factor; surgery for endometriosis, not surgery in endometriosis.

Response: We already amended as your suggestion.

Previous manuscript: Line 3 ….. “Risk factor” ……. “surgery in endometriosis”

Revised manuscript: Line 3 …… “Risk factors” ……. “surgery for endometriosis”

Line 31 (abstract): with the same or higher VAS; add "or higher".

Response: We already amended as your suggestion.

Previous manuscript: Line 31 ….. “with the same”

Revised manuscript: Line 31 …… “with the same or higher”

Line 58 (intro): glands, not gland.

Response: We already amended as your suggestion.

Previous manuscript: Line 58 ….. “an endometrial gland”

Revised manuscript: Line 58 …… “an endometrial glands”

Line 69 (intro): add some reference on the general treatment of endometriomas, such as Chapron C et al "Management of ovarian endometriomas", Hum Reprod Update 2002, and Muzii L et al "Management of endometriomas", Semin Reprod Med 2017.

Response: We already amended as your suggestion.

Previous manuscript: Line 69 ….. “[4]”

Revised manuscript: Line 70…… “[5–7]”

 Line 335-340 ……“6. Chapron C, Vercellini P, Barakat H, Vieira M, Dubuisson JB. Management of ovarian endometriomas. Hum Reprod Update. 2002; 8(6): 591-597. https://doi.org/doi: 10.1093/humupd/8.6.591 PMID: 12498427

7. Muzii L, Di Tucci C, Di Feliciantonio M, Galati G, Verrelli L, Donato VD, et al. Management of Endometriomas. Semin Reprod Med. 2017; 35(1): 25-30. doi: https://doi.org/10.1055/s-0036-1597126 PMID: 27926971”

Line 81 (intro): too many references to attest the high endometriosis load of the Authors (refs 17, 18, 19). Select only one of the three. Ref 20 is on the other hand appropriate in the text.

Response: We already amended and delete the reference No. 17-19 and selected only reference No. 21 as your suggestion.

Previous manuscript: Line 81-82 ….. “years [17–19].A previous study by Cheewadhanaraks et al. (2013) [20] in our hospital showed that”

 Line 359-369…. “17. Cheewadhanaraks S, Choksuchat C, Wattanakumtornkul S. Estrogen plus progestin versus estrogen after definitive surgery for endometriosis: a study of pain recurrence. J Med Assoc Thai. 2013; 96(8): 881–887. PMID: 23991592

18. Cheewadhanaraks S, Choksuchat C, Dhanaworavibul K, Liabsuetrakul T. Postoperative depot medroxyprogesterone acetate versus continuous oral contraceptive pills in the treatment of endometriosis-associated pain: a randomized comparative trial. Gynecol Obstet Invest. 2012; 74(2): 151–156. https://doi.org/10.1159/000337713 PMID: 22722530

19. Cheewadhanaraks S, Peeyananjarassri K, Choksuchat C, Dhanaworavibul K, Choobun T, Bunyapipat S. Interval of injections of intramuscular depot medroxyprogesterone acetate in the long-term treatment of endometriosis-associated pain: a randomized comparative trial. Gynecol Obstet Invest. 2009; 68(2): 116–121. https://doi.org/10.1159/000226090 PMID: 19556801”

Revised manuscript: Line 83-86 ……. “years and our previous study showed that the cumulative recurrence rates of pain after depot medroxyprogesterone acetate (DMPA) treatment at months 12, 24, 36, 48, and 60 were 18%, 28%, 41%, 46%, and 50%, respectively [20].”

Line 264 (discussion): close follow-up, not closed

Response: We already amended as your suggestion.

Previous manuscript: Line 264 ….. “closed”

Revised manuscript: Line 264 …… “close”

Reviewer #2: Dear author, thank you for the opportunity to analyze this interesting study on such a widespread pathology.

However, I believe that some elements should be specified in the introduction:

- endometriosis can be responsible for symptoms even in the menopausal age, varying between 2 and 5% of cases [you can cite: M M Ianieri et al. Retroperitoneal endometriosis in postmenopausal woman causing deep vein thrombosis: case report and review of the literature]

Response: We already amended as your suggestion.

Previous manuscript: -

Revised manuscript: Line 60-61 ……“Uncommonly, it was found to vary between 2% and 5% in menopausal women [3].”

Line 326-328….. “3. Ianieri MM, Buca DIP, Panaccio P, Cieri M, Francomano F, Liberati M. Retroperitoneal endometriosis in postmenopausal woman causing deep vein thrombosis: case report and review of the literature. Clin Exp Obstet Gynecol. 2017; 44(1): 148-150. PMID: 29714887.”

- I suggest specifying that the recurrence rate of endometriosis also varies according to the definition of recurrence among the various authors

Response: We already amended as your suggestion.

Previous manuscript: Line 70-71 ….. “The 2-year and overall recurrence rates of endometriosis after conservative surgery were 9–29% and 13–30%, respectively [5–8].

Revised manuscript: Line 71-73 ……“ The recurrence rates of endometriosis after conservative surgery varied according to the definition of recurrence and were reported as 9–29% and 13–30% at 2-year and overall recurrence rates, respectively [8–11].”

In the discussion session:

- advice to improve and enrich the discussion about the potential role of deep endometriosis on disease recurrence, particularly if radical surgery has not been performed. It would in fact be suggested to specify this aspect in the results or even in the tables. 

In particular, it would be interesting to know in your series the incidence of endometriosis of the parameters which can often be associated with ovarian endometriosis and whose non-treatment can be associated with persistence/recurrence of symptoms [Ianieri MM et al. Impact of nerve-sparing posterolateral parametrial excision for deep infiltrating endometriosis on postoperative bowel, urinary, and sexual function].

You cannot speak of ovarian recurrence without specifying the possible infiltration of the parameters which are anatomical structures adjacent to the ovaries

Response: We already amended as your suggestion.

Previous manuscript: -

Revised manuscript: Line 264-275 ……“In the present study, endometrioma was the most common finding (86.7%), co-existing with the presence of obliterate cul-de-sac (75.4%), uterosacral ligament involvement (53.2%), and deep endometriosis (23.2%). Previous studies showed higher incidences of co-existing uterosacral ligament involvement (73.8%) and deep endometriosis (43.2%) [31, 32]. These discrepancies may be attributed to the study design, which had missing data (64% and 81% in uterosacral ligament involvement and deep endometriosis, respectively) and all of the operative finding scores based on rAFS classification. Although rAFS classification is accepted and correlates to surgical complexity, Enzian classification is suitable for defining deep endometriosis [33]. Lanieri et al. (2022) [34] demonstrated that parametrectomy significantly improved the symptoms of dyschezia, dysmenorrhea, dysuria, dyspareunia, and chronic pelvic pain but worsened female sexual dysfunction. The failure to detect deep endometriosis resulted in undertreatment and, ultimately, a lack of improvement in their symptoms.”

Line 430-443….. “31. Hajati A, Hajati O. A review of more than 2000 cases of site-specific pelvic endometriosis rates by MRI: a guide to minimizing under/overdiagnosis non-invasively. Insights Imaging 2022; 13(1): 129. https://doi.org/10.1186/s13244-022-01270-z PMID: 35939136

32. Piriyev E, Schiermeier S, Römer T. Coexistence of endometriomas with extraovarian endometriosis and adhesions. Eur J Obstet Gynecol Reprod Biol. 2021; 263: 20–24. https://doi.org/10.1016/j.ejogrb.2021.05.044 PMID: 34144489

33. Hudelist G, Valentin L, Saridogan E, Condous G, Malzoni M, Roman H, et al. What to choose and why to use - a critical review on the clinical relevance of rASRM, EFI and Enzian classifications of endometriosis. Facts Views Vis Obgyn. 2021; 13(4): 331–338. https://doi.org/10.52054/FVVO.13.4.041 PMID: 35026095

34. Ianieri MM, Raimondo D, Rosati A, Cocchi L, Trozzi R, Maletta M, et al. Impact of nerve-sparing posterolateral parametrial excision for deep infiltrating endometriosis on postoperative bowel, urinary, and sexual function. Int J Gynaecol Obstet. 2022; 159(1): 152–159. https://doi.org/10.1002/ijgo.14089. PMID: 34995374”

6. PLOS authors have the option to publish the peer review history of their article (what does this mean?). If published, this will include your full peer review and any attached files.

Do you want your identity to be public for this peer review? For information about this choice, including consent withdrawal, please see our Privacy Policy.

Reviewer #1: No

Reviewer #2: No

Response: Thank you for your thoughtful comments.

Additionally, we have some changes about the 2-year recurrence rate of endometriosis; the percentage is incorrect. We have already rechecked and amended.

Previous manuscript: Line 39, 45, 167, 202, 274…….. “24.3%”

 Line 167……. “88”

Revised manuscript: Line 39, 45, 169, 204, 294….. “23.2%”

 Line 169……. “84”

Finally, because two references were included in this revised manuscript (references no. 5 and 6), the next references were re-arranged as follows:

 Previous manuscript Revised manuscript

Reference No. 3 4

 4 5

 5 8

 6 9 

 7 10

 8 11

 9 12

 10 13

 11 14

 12 15

 13 16

 14 17

 15 18

 16 19

 17 delete

 18 delete

 19 delete

---

## [Editor Report · Decision Letter 1]

27 Jul 2023

Risk factors for recurrent endometriosis after conservative surgery in a quaternary care center in southern Thailand

PONE-D-22-29483R1

Dear Dr. Klangsin,

We’re pleased to inform you that your manuscript has been judged scientifically suitable for publication and will be formally accepted for publication once it meets all outstanding technical requirements.

Kind regards,

Diego Raimondo

Academic Editor

PLOS ONE
---

## [Editor Report · Acceptance letter]

1 Aug 2023

PONE-D-22-29483R1 

Risk factors for recurrent endometriosis after conservative surgery in a quaternary care center in southern Thailand 

Dear Dr. Klangsin:

I'm pleased to inform you that your manuscript has been deemed suitable for publication in PLOS ONE. Congratulations! Your manuscript is now with our production department. 

Kind regards, 

on behalf of

Dr. Diego Raimondo 

Academic Editor

PLOS ONE